# Tinnitus treatment by vagus nerve stimulation: A systematic review

I. Stegeman[1,2]*, H. M. Velde[1,2], P. A. J. T. Robe[2,3], R. J. Stokroos[1,2], A. L. Smit[1,2]

**1** Department of Otorhinolaryngology, Head and Neck Surgery, University Medical Center Utrecht, Utrecht, Netherlands, **2** University Medical Center Utrecht Brain Center, University Medical Center Utrecht, Utrecht, Netherlands, **3** Department of Neurology and Neurosurgery, University Medical Center Utrecht, Utrecht, Netherlands

* I.Stegeman@umcutrecht.nl

## Abstract

### Background

Tinnitus is a phantom sensation of sound, which can have a negative impact on quality of life of those affected. No curative treatments are currently known. Neuromodulation by vagus nerve stimulation has emerged as a new treatment option for tinnitus, though till date the effectiveness remains unclear. Therefore, we aim to review the effect of vagus nerve stimulation on tinnitus distress and tinnitus symptom severity in patients with chronic tinnitus.

### Methods

We searched Pubmed, Embase and the Cochrane Library systematically for RCTs, observational studies and case studies on the effect of VNS treatment for tinnitus on October 29, 2019. Studies including adult patients with subjective tinnitus, comparing transcutaneous or implantable VNS to placebo or no treatment or before and after application of VNS treatment on tinnitus distress and tinnitus symptom severity measured with a validated questionnaire were eligible. The risk of bias was assessed with the appropriate tool for each type of study.

### Results

Our search identified 9 primary studies of which 2 RCTs, 5 cohort studies and 2 case series or reports. 5 studies used transcutaneous VNS treatment and 4 used implanted VNS treatment. 6 studies combined VNS treatment with sound therapy. There was a serious risk of bias in all studies, especially on confounding. Most studies reported a small decrease in tinnitus distress or tinnitus symptom severity.

### Conclusion

Due to methodological limitations and low reporting quality of the included studies, the effect of VNS on tinnitus remains unclear. To draw conclusions for which patient population and to what extent (t)VNS is beneficial in the treatment of tinnitus, a randomised controlled trial should be considered.

**Data Availability Statement:** All relevant data are within the manuscript.

**Funding:** The author(s) received no specific funding for this work.

**Competing interests:** The authors have declared that no competing interests exist.

## Introduction

Subjective tinnitus is considered a phantom sensation of sound, experienced in the absence of any internal or external acoustic stimulus [1]. At present, the prevalence of tinnitus is unclear with published figures varying from 5.1 to 42.7% [2]. Not only tinnitus itself but also associated depressive symptoms [3], anxiety [4], and insomnia [5] can have a negative impact on quality of life of those affected. It is estimated that in about 20% of adults who experience tinnitus, clinical intervention is required [6]. Currently, no curative evidence based treatments for tinnitus exist. Considering the effects of tinnitus on daily life, an effective and safe therapy for tinnitus is of considerable importance to reach out to this unmet clinical need.

Vagus nerve stimulation (VNS) is approved by the Food and Drug Administration (FDA) for both refractory epilepsy and resistant depression. Neuromodulation by VNS has emerged as a new treatment option for tinnitus. The underlying mechanism involves activating the nucleus of the solitary tract, which in turn can activate the locus coeruleus and nucleus basalis, which then release neuromodulators that have effects on plasticity regulation by modulating neurons in the cortex [7]. As tinnitus is associated with an imbalance between excitation and inhibition that may lead to map reorganization and increased synchronous firing of auditory neurons [8], neuromodulation by VNS might have a positive influence on this process. This idea was supported by a study by Engineer et al. in which the application of VNS in combination with pairing tones significantly increased excitability, suppressed spontaneous multi-unit activity in the auditory cortex, and completely eliminated tinnitus in noise-exposed rats [9].

Traditionally, VNS is performed by implantation of a device connected to an electrode located along the cervical branch of the vagus nerve. Meanwhile, a non-invasive transcutaneous device (tVNS) has been developed to stimulate the auditory branch of the vagus nerve (ABVN), which demonstrated to result in similar functional MRI features with changes of brain activation compared to invasive VNS [10]. VNS to relief tinnitus symptoms has also been applied in combination with other interventions, such as sound stimuli, to increase frequency selectivity and decrease cortical synchronization [11].

Considering the lack of knowledge about the effectiveness of VNS, we aim to systematically review the effect of vagus nerve stimulation on tinnitus distress and tinnitus symptom severity in patients with tinnitus.

## Methods

The PRISMA 2009 statement was used as a guideline for writing this review [12].

### Eligibility criteria

The following study characteristics were used for including primary studies on the effect of VNS treatment for tinnitus.

Participants: adult patients with subjective tinnitus. No restriction for level of tinnitus perception or tinnitus duration was made. Animal studies were excluded from this review.

Interventions; transcutaneous or implantable VNS.

Comparison: placebo or no treatment or comparison before and after application of VNS treatment.

Outcome: Tinnitus symptom severity (e.g. perceived tinnitus severity, impact of tinnitus on patient's life, tinnitus related handicap, measured with tinnitus loudness and annoyance visual analogue scales (VAS, range 0–10 unless indicated otherwise) or validated questionnaires and tinnitus distress (e.g. psychological aspects of tinnitus complaint and distress including depression or anxiety) as measured by a validated distress questionnaire, as well as adverse events due to VNS implantation or application. Included validated questionnaires are: the Tinnitus

Handicap Inventory (THI, range 0–100), the Tinnitus Questionnaire (TQ, range 0–82), the mini-TQ (range 0–24) the Tinnitus Severity Scale (15 items), the Tinnitus Handicap Questionnaire (THQ, range 0–100), the Tinnitus Functional Index (TFI, range 0–100), the Tinnitus Reaction Questionnaire (TRQ, range 0–104),

Due to the early nature of the VNS intervention for tinnitus, RCTs, observational studies and case studies will be included in this study. No language, date, or publication status restriction are used in the search.

## Literature search

We performed a systematic search in Medline, Embase and the Cochrane Library databases on October 29, 2019. Protocols for ongoing trials were searched via www.clinicaltrials.gov. In case of ongoing trials, investigators were contacted for information. The search syntax is shown in Table 1.

## Selection of studies

After removal of duplicates, title and abstract screening was performed independently by two authors (IS or HV and ALS) according to predetermined inclusion and exclusion criteria. Eligible full text articles were retrieved through the databases and by emailing authors. Subsequently, full texts of eligible articles were screened independently (IS or HV and ALS). Cross reference checking of included studies was used. Discordances regarding inclusion were resolved by discussion and consensus.

## Quality assessment

The risk of bias of non-randomized studies was assessed by use of the ROBINS-1 tool [13]. The Cochrane Risk of Bias tool was used for the quality assessment of RCTs [14]. No studies were excluded based on risk of bias, as advised by the Cochrane risk of bias group [14].

## Data extraction and analysis

Two researchers (IS or HV and ALS) independently extracted descriptive data regarding type of study, baseline and inclusion criteria of participants, tinnitus duration, studied intervention, outcome data, and follow up duration.

Because of the expected heterogeneity of studies in methodology, inclusion criteria of participants, and assessed outcomes, the intentional analysis is a descriptive synthesis of results and not a meta-analysis.

**Table 1. Search syntax.**

Medline
((((((tinnitus[MeSH Terms]) OR tinnitus[Title/Abstract])) AND (((((vagal nerve[Title/Abstract]) OR vagus nerve [Title/Abstract]) OR vagus nerve[MeSH Terms]) OR nervus X[Title/Abstract]) OR tenth cranial nerve[Title/ Abstract] OR VNS [Title/Abstract]))))
Embase
tinnitus:ab,ti AND ('vagus nerve':ab,ti OR vagal:ab,ti AND nerve:ab,ti OR (nervus:ab,ti AND x:ab,ti) OR (tenth:ab,ti AND cranial:ab,ti AND nerve:ab,ti) OR vns:ab,ti)
Cochrane Library
tinnitus:ab,ti,key AND ('vagus nerve':ab,ti,key OR vagal nerve:ab,ti,key OR nervus vagus:ab,ti,key, tenth cranial nerve: ab,ti, key OR vns:ab,ti,key)
ClinicalTrials.gov
Vagus Nerve | Tinnitus

## Results

### Literature search and selection

We found 59 studies on Medline, 74 on Embase and 15 on the Cochrane Library. This resulted in a total of 100 studies after removal of duplicates. Title and abstract screening resulted in 40 studies eligible for full text screening (Fig 1). Thereof, 10 primary studies or protocols met the inclusion criteria, of which one study was excluded from further analyses because it presented outcome which was also described as part of a larger cohort in a manuscript by Tyler et al. [15].

### Study characteristics

No pooling of results was possible, since studies were heterogeneous in methodology, inclusion of participants and assessed outcomes. Characteristics of included studies are shown in Table 2.

Of the 9 included primary studies, 5 used transcutaneous VNS treatment [16–20] compared to 4 studies with implanted VNS [21–24]. In 6 out of 9 studies the VNS treatment was combined with sound therapy(ST) [17, 18, 20, 22–24].

Suk et al. performed a prospective cohort study in 2018 in which 24 patients with a tinnitus duration of ≥3 months received four sessions of transcutaneous VNS over a two week period [16]. In these sessions, the cavum, cymba, and tragus were subsequently stimulated to the maximal sensory thresholds, this being the threshold that could be tolerated without any painful sensation. The outcome of their study was tinnitus symptom severity measured by a VAS regarding tinnitus loudness, awareness, annoyance and its effect on life, as well as THI scores one month after treatment.

Wichova et al. conducted a study of the effect of implanted VNS on perception of tinnitus in epilepsy patients [21]. 20 patients who received a VNS implant as a treatment for refractory epilepsy and with pre-operative tinnitus were tracked down retrospectively by chart research and phone surveys. The mean VNS use of included patients was 3.3 years at time of evaluation. A VAS of tinnitus loudness was used to address pre- and post-operative tinnitus retrospectively.

Tyler et al. published a double-blinded pilot RCT in 2017 in which 30 patients were implanted with a vagus nerve stimulator to explore the safety and efficacy of VNS therapy [22]. In the first 6 weeks, daily stimulation with paired tones was compared to a control group receiving VNS stimulation with unpaired tones. In the second 6 weeks the control group crossed-over to VNS stimulation with paired tones similar to the intervention group, where after participants were followed up to one year, and outcome for both groups were pooled based on THI, THQ, TFQ and tinnitus loudness scale.

De Ridder et al. completed a case study in 2015 describing one patient with chronic tinnitus [23]. After VNS implantation, daily vagus nerve stimulation was applied paired with tones for 4 weeks. The patient was followed up for two months after end of stimulation and tinnitus distress and symptom severity was assessed by the THI, TRQ and the THQ.

Shim et al. included 30 patients in 2015 with chronic tinnitus for more than 12 months who were unresponsive to therapy [17]. Transcutaneous VNS was applied by a patch in the auricular concha paired with notched music for 10 sessions. A VAS of tinnitus loudness and the THI were measured before and direct after treatment.

Mei et al. conducted a RCT comparing a 8-week daily treatment of conchal acupoint tVNS to oral drug therapy with flunarizine hydrochloride and oryzanol [18]. They included tinnitus patients visiting an outpatient department as well as volunteers with tinnitus from university

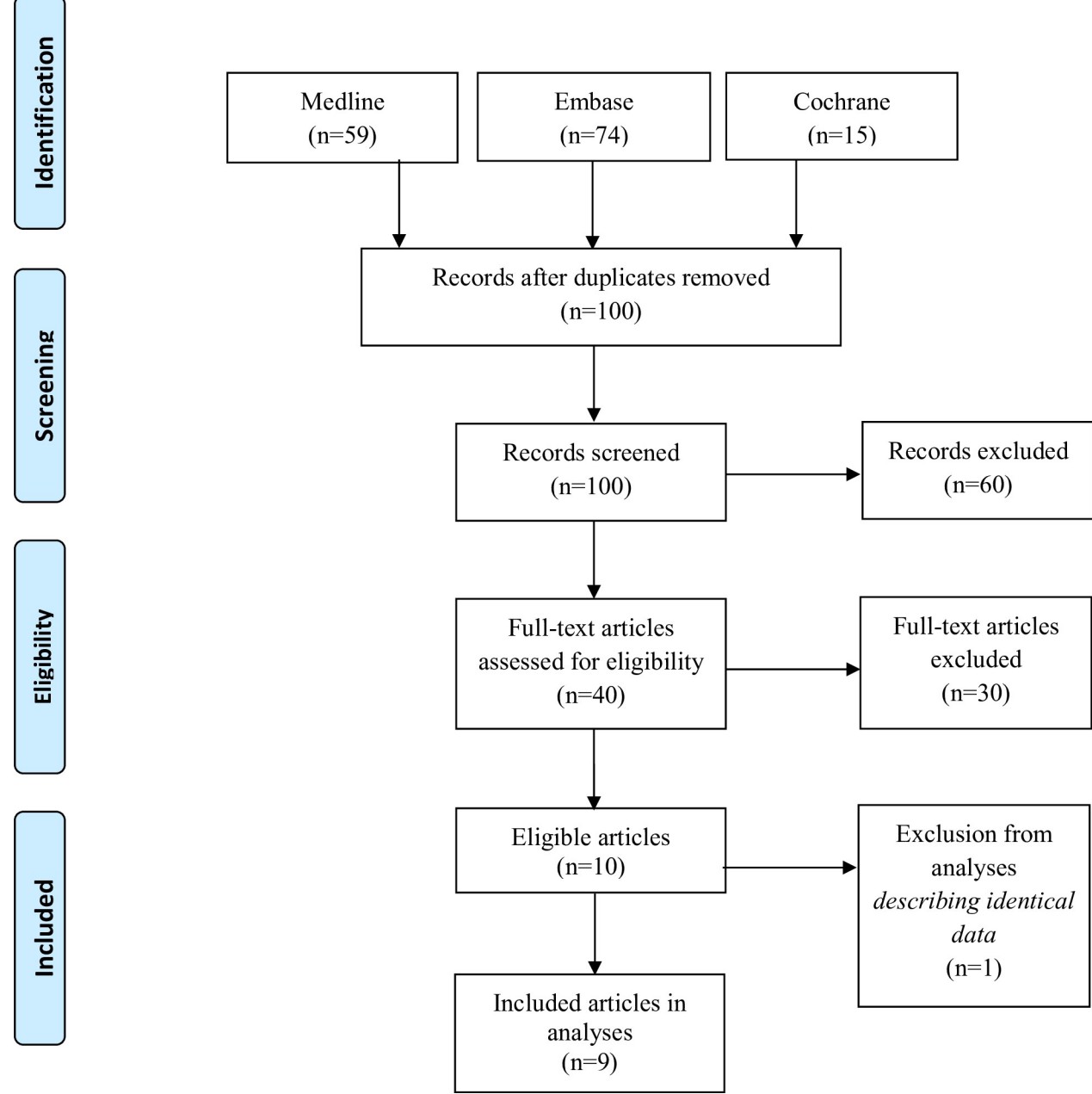

**Fig 1. PRISMA 2009 flow diagram.**

with different durations of tinnitus (recurrent tinnitus >1 month or sustained > 5 days). The outcome of tinnitus symptom severity in their study was measured by THI per ear resulting in numbers of ears within THI score levels (1 to 5) before and after treatment per group.

**Table 2. Study characteristics.**

| Study | Study type | Mean tinnitus duration (range)˚ | Inclusion criteria | Intervention | Outcome | Time of outcome assessment |
|---|---|---|---|---|---|---|
| **VNS without paired sound** | | | | | | |
| **Suk, 2018** | Cohort, prospective (n = 24) | 31 m ± 49 (3–204) | Tinnitus duration > 3 months | tVNS | THI, VAS (loudness, awareness, annoyance, effect on life) | 1 month after end of tVNS sessions |
| **Wichova, 2018** | Cohort, retrospective (n = 20) | X | Pre-operative tinnitus | Implanted VNS | VAS (loudness) | After 3.3 (± 2.0) years of VNS use |
| **Kreuzer, 2014** | Cohort, prospective (n = 50) | X | Tinnitus duration ≥ 6 months, TQ ≥ 31 | tVNS afferent auricular branch | TQ, THI, rating scale of tinnitus loudness, annoyance, discomfort, distractibility, unpleasentness (range not reported) | Phase 1: at the end of 45.5±21.0 days of stimulation |
| | | | | | | Phase 2: at the end of 24 weeks of stimulation and 4 weeks after end of stimulation |
| **VNS with paired sound** | | | | | | |
| **Tyler, 2017** | RCT, cross-over (n = 30) | Intervention: 18.8 ± 17.1 y | Chronic tinnitus, unresponsive to therapy, BDI < 30 | Intervention group: implanted VNS plus paired tones | THI, THQ, TFI, VAS$ (loudness severity) | At the end of 6 and 12 weeks and after 1 year of daily stimulation |
| | | Control: 10.1 ± 10.3 y | | Control group: implanted VNS plus unpaired tones | | |
| **De Ridder, 2015** | Case Report (n = 1) | 59 m | Chronic tinnitus, unresponsive to therapy | Implanted VNS, paired tones | THI, TRQ, THQ, | At the end of 4 weeks of stimulation and 2 months after end of stimulation |
| **Shim, 2015** | Cohort, prospective (n = 30) | 42.40 m ± 45.64 (12–240) | Chronic tinnitus > 12 months, unresponsive to therapy | tVNS auricular concha plus notched music | THI, VAS (loudness) | At the end of 10 treatment sessions |
| **Mei, 2014** | RCT (n = 63) | X | Recurrent tinnitus > 1 month or recurrent tinnitus > 5 days | Intervention group: tVNS auricular concha acupoints plus tones | THI¥ | At the end of 4 and 8 weeks of stimulation |
| | | | | Control group: oral flunarizine hydrochloride + oryzanol | | |
| **De Ridder, 2014** | Case Series, prospective (n = 10) | X | Chronic tinnitus, TRQ ≥18, unresponsive to therapy | Implanted VNS, paired tones | THI, TRQ, THQ | After 4 weeks of stimulation and 3 to 6 months after end of stimulation |
| **Lehtimäki 2013** | Cohort, prospective (n = 10) | X | Disturbing tinnitus | tVNS tragus plus sound therapy | THI, mini-TQ, VAS (loudness, annoyance) | After 7 treatment sessions |

˚ outcome reported in months (m) or years (y); X = not reported; tVNS = transcutaneous VNS; ¥ THI score per tinnitus ear; BDI = Beck Depression Inventory; $ VAS range 0–100.

De Ridder et al. used the same methods as in 2015 for their 2014 study on 10 patients with chronic tinnitus with a duration of ≥ 1 year and a TRQ score ≥ 18 which were unresponsive to previous therapy [24]. With similar outcome measures they followed these patients 3–6 months after the end of treatment to measure tinnitus distress and symptom severity.

Kreuzer et al. included a cohort of 24 patient with a tinnitus duration ≥6 months and a TQ score of ≥31 points, in which tVNS of the afferent auricular branch was used with daily stimulation for 24 weeks [19]. The first phase of the study, including 24 patients, was terminated

**Table 3. Risk of bias according to the ROBINS-I tool.**

| Study | Confounding | Selection of participants | Classification of intervention | Deviations from intervention | Missing Data | Measurement of outcomes |
|-------|-------------|---------------------------|-------------------------------|------------------------------|--------------|-------------------------|
| Suk, 2018 | Serious | Low | Low | Low | Low | Low |
| Wichova, 2018 | Serious | Serious | Unclear | Unclear | Serious | Low |
| De Ridder, 2015 | Serious | Serious | Low | Serious | NA | Low |
| Shim, 2015 | Serious | Low | Low | Low | Low | Low |
| Kreuzer, 2014 | Serious | Serious | Low | Serious | Moderate | Low |
| De Ridder, 2014 | Serious | Low | Low | Low | Moderate | Low |
| Lehtimäki, 2013 | Critical | Unclear | Low | Low | Low | Unclear |

after two cardiac adverse events, and started again with an improved stimulation device resulting in treatment of another 24 patients in phase 2. In total 16 patients dropped out of the study in the two phases (10/24 in phase 1, 6/24 in phase 2). TQ, THI, tinnitus loudness and annoyance scales were assessed at baseline, end of stimulation and 4 weeks thereafter.

Lehtimäki et al. included 10 patients in a pilot study in which tVNS was delivered to the tragus in combination with ST [20]. Tinnitus loudness and annoyance scales, THI and the mini-TQ were assessed before and after 7 stimulation sessions during 10 days.

## Risk of bias assessment

The risk of confounding was serious or critical in all included studies (Tables 3 and 4). The risk of selection bias of participants was serious in three studies [19, 21, 23], in which patients were not included in a consecutive manner. Interventions were well defined. Deviations from the protocol were serious in two studies [19, 23]. In 1 study there were serious concerns on missing data [21]. The risk of bias on outcome measures was low in all but one study, in which it was unclear [20]. The study of Wichova et al. had a high risk on recall bias due to their retrospective study design [21]. Methods for randomization were not described in both RCT's [18, 22]. Tyler first blinded and then deblinded their participants which might have influenced their long term outcomes [22].

## Outcomes

**Tinnitus questionnaires.** Table 5 shows the outcomes of the included studies. Six out of nine studies performed statistical analyses in their studies [16–19, 21, 22]. Suk et al., found a significant decrease of the mean THI from 45 (± 19) before and 27 (± 15) after treatment (p < 0.001) [16]. Also, they found that all VAS scores (tinnitus loudness, awareness, annoyance, effect on life) improved after treatment (p < 0.05), although they did not report any

**Table 4. Risk of bias according to Cochrane risk of bias tool.**

| Study | Random sequence generation | Allocation concealment | Blinding of participants and personnel | Blinding of outcome assessor | Incomplete data | Selective reporting |
|-------|----------------------------|------------------------|----------------------------------------|------------------------------|-----------------|---------------------|
| Tyler, 2017 | ? | ? | + | + | - | - |
| Mei, 2014 | ? | ? | ? | ? | ? | ? |

? Unclear risk of bias;—low risk of bias; + high risk of bias.

**Table 5. Outcomes.**

| Study | Follow up | THI | TRQ | THQ | TQ | TFI | Annoyance | Loudness |
|---|---|---|---|---|---|---|---|---|
| **Without sound pairing** | | | | | | | | |
| **Suk, 2018** | Pre-treatment | 45 (± 19) | x | x | x | x | x | x |
| | Immediately after treatment | 27 (± 15) | x | x | x | x | 33.3% (n = 8) responders | 45.8% (n = 11) responders |
| **Wichova, 2018** | Pre-operative | x | x | x | x | x | x | 5.85 (± 1.5) |
| | Post-operative | 41.75 (± 22.6) | x | x | x | x | x | 3.8 (± 1.4) |
| **Kreuzer, 2014** | Pre-treatment | 50.0 (± 19.4) | x | x | 48.8 (± 11.7) | | 6.8 (± 2.1) | 6.8 (± 1.9) |
| **phase 1** | Immediately after treatment | 49.4 (± 21.8)* | x | x | 45.2 (±14.8)# | | 7.1 (± 2.2)* | 6.9 (± 1.9)* |
| With sound pairing | | | | | | | | |
| **Tyler, 2017** | Immediately after treatment ∞ | Intervention: -17.7 (-28 to -7.3)# | x | Intervention: -2.5 (-8.3 to -3.3) | x | Intervention: -2.03 (-7.1 to 3.1) | x | Intervention: -6.69 (-13.26 to -0.11) |
| | | Control: -7.3 (-27.5 to 12.7) * | | Control: -7.5 (-15.8 to 0.7) | | Control: -7.5 (-15.5 to 0.7) | | Control: -8.5 (-22.6 to 5.5) |
| | End of follow-up £ | Intervention: -19.39 (-37.99, -0.79) # | x | Intervention: -11.99 (-19.72, -4.26) # | x | Intervention: -9.98 (-19.74, -0.21)# | x | Intervention: -19.41 (-34.01, -4.82) # |
| **De Ridder, 2015** | Pre-treatment | 50 | 55 | 63 | x | | x | x |
| | Immediately after treatment | 26 | 17 | 40 | x | | x | x |
| | End follow-up | 21 | 40 | 45 | x | | x | x |
| **Shim, 2015** | Pre-treatment | 41.50 (± 29.64) | x | x | x | | x | 6.32 (± 2.06) |
| | Immediately after treatment | 35.46 (± 23.30)* | x | x | x | | x | 5.16 (± 1.52) # |
| **Kreuzer, 2014** | Pre-treatment | 55.3 (± 21.3) | x | x | 52.6 (±14.8) | | 7.0 (± 2.0) | 7.0 (± 1.6) |
| **phase 2** | Immediately after treatment | 58.0 (± 25.3) | x | x | 49.8 (± 19.0) | | 7.4 (± 2.1) | 7.3 (± 2.0) |
| | End of follow-up | 57.0 (± 26.2)* | x | x | 49.8 (±19.5)* | | 7.3 (± 2.2)* | 7.1 (± 2.0)* |
| **Mei, 2014¥** | Pre-treatment | *Score 0–36*: VNS 13/50 | x | x | x | | x | x |
| | | Control 13/46 | | | | | | |
| | | *Score 38–100*: VNS 37/50 | | | | | | |
| | | Control 33/46 | | | | | | |
| | Immediately after treatment | *Score 0–36*: VNS 38/50 # | x | x | x | | x | x |
| | | Control 21/46 | | | | | | |
| | | *Score 38–100*: VNS 12/50# | | | | | | |
| | | Control 25/46* | | | | | | |
| **De Ridder, 2014** | Pre-treatment | 69.6 | 66.6 | 72.0 | x | | x | x |
| | Immediately after treatment | 61.4 | 56.0 | 67.3 | x | | x | x |
| | End follow-up | 62.0 | 56.0 | 68.5 | x | | x | x |
| **Lehtimäki, 2013** | Pre-treatment | 70 | x | x | 12† | | 61 | 59 |
| | Immediately after treatment | 39 | x | x | 10† | | 40 | 38 |

Numbers present outcome measurement with standard deviations in brackets when given.

x = not reported

* = not statistically significant

# = statistically significant

† = mini-TQ

¥ = reporting numbers of tinnitus ears per group within outcome scores of an individual test; ∞ = scores reported in change from baseline, in brackets 95% confidence intervals

£ = scores reported in change from baseline, in brackets: lower CI, upper CI, n participants.

figures on VAS scores before and after treatment. They dichotomized the VAS outcomes to responders, with a decrease of at least 50%, and non-responders. The response rates were 33.3% (n = 8), 62.5% (n = 15), 45.8% (n = 11), and 41.7% (n = 10) on respectively tinnitus loudness, awareness, annoyance, and its effect on life. In the retrospective study of Wichova et al., all patients with pre-operative tinnitus continued to have tinnitus after VNS implantation [21]. In 4 out of 20 patients the VAS loudness did not change, in 16 out of 20 there was at least 1 point decrease on a 10 point scale. The mean difference was 2.1 ± 1.8 and statistically significant (p < 0.001). The RCT by Tyler et al., using an implanted VNS, showed that after 6 weeks paired stimulation the THI score improved compared to the control group of unpaired VNS, with a between-group difference of 10.3% which was not statistically significant (p = 0.3393) [22]. Secondly, they reported a clinically meaningful response (defined as 7-point cut-off for the THI) in 50% of participants after one year of paired VNS usage. The other studies included in our review using statistical analyses of their outcomes used tVNS with [17, 18] or without [19] sound therapy. Of the studies including sound therapy, Mei et al. reported THI level reduction after tVNS treatment plus tones compared to a control population with oral drugs [18], where in the study by Shim et al. tVNS treatment plus notched music demonstrated no significant change of the THI compared to the scores pre-stimulation [17]. However, the latter showed a small statistically significant decrease in the tinnitus loudness immediately after end of treatment (pre-treatment 6.32 ± 2.06 vs. after treatment 5.16 ± 1.52). Kreuzer et al. [25] demonstrated a small significant decrease in TQ after tVNS treatment in the first phase of their study, which was terminated prematurely (TQ pre-treatment 48.8 ± 11.7 vs after treatment 45.2 ± 14.8) [19]. No significant decrease in TQ or annoyance and loudness was observed in the second phase. Included studies using descriptive analyses by De Ridder et al. and Lethimäki et al., showed a small decrease in tinnitus symptom severity and tinnitus annoyance and loudness scales after vagus nerve stimulation compared to before treatment [20, 23, 24]. These decreases were not statistically tested.

**Harms.** Several studies reported on adverse effects of the VNS implantation [16, 19–24]. Tyler et al. reported that two out of 30 patients experienced hoarseness after VNS implantation [22], Wichova et al., reported a similar outcome in 1 out of 56 interviewed patients [21], and also the patient of the case-report by De Ridder et al., reported hoarseness and a transient left vocal cord hypomobility and slight inflammation of the abdominal surgical site [23]. De Ridder et al., do not report figures on side-effects in their case series [24]. In the cohort of 50 patients of Kreuzer et al., 50 adverse events were reported, e.g. dysesthesia, skin redness, hoarseness and arrhythmia [19] Suk et al. and Lehtimäki et al. described that none of the patients experienced adverse effects [16, 20]. The other studies did not include reporting of complications or harms by the intervention as an outcome of the study in their methods or in the results [17, 18].

## Discussion

So far, the majority of applied tinnitus therapies (e.g. hearing aids, active sound therapy, pharmacotherapy) have been demonstrated to be insufficient to reverse the pathological changes that cause tinnitus. Recently, different neuromodulation techniques have been developed to treat tinnitus (e.g. transcranial magnetic stimulation or direct current stimulation or implantation of electrodes at different sites). In this systematic review we aimed to verify whether neuromodulation by vagus nerve stimulation is effective in relieving tinnitus complaints.

We conducted a systematic search resulting in the inclusion of nine studies in our analyses that reported primary data on the effect of (t)VNS with or without sound therapy. Included studies are heterogeneous in methods, inclusion of participants, and assessed outcomes and

have important methodological limitations: considerable risk of bias, low sample sizes, and limitations in quality of writing of the methodology which makes replication hard. Considering these limitations, most studies reported a small decrease in tinnitus distress or symptom severity immediately after treatment, using different modalities. Four studies reported outcomes after follow-up of several months [19, 24] or up to one [22] or over three years [21]. However, given the aforementioned limitations, no strong conclusions could be drawn from these data on the effect of (t)VNS with or without sound therapy, nor about differences between implanted or transcutaneous VNS. This stresses the need for improvements in the quality and reporting of (tinnitus) research, and provide homogeneity in outcomes. The great efforts being made by tinnitus researchers and clinicians to find treatment options for our patients will have more impact if we diminish bias, use optimal research designs and improve quality of reporting of our studies.

The change in tinnitus outcomes was small in most studies, and one could question the clinical relevance of these decreases. It might be possible that some subgroups of tinnitus patients have greater benefit by VNS, with or without sound therapy, than others. An effect in subgroups was found in the study of Tyler et al. with greater benefit for participants with tonal and non-blast induced tinnitus [22] and in the study by Shim et al., in which a shorter duration of tinnitus complaints was more favourable to benefit from VNS treatment [17]. Though, both studies were underpowered for these specific analyses. Evidence concerning VNS for tinnitus is scarce. In 2015 a protocol for a systematic review has been published on neuromodulation for tinnitus at Cochrane, which is including randomized controlled trials of VNS. Because of the early status of the intervention, we believe not limiting our review to RCT's provides us with all the information about VNS so far [26].

The differences in applied VNS techniques can be of importance for the effect of the treatment. Theoretically, stimulation of the auricular branch of the vagus nerve might be less effective than cervical VNS application because fibres of the auricular branch only partly target the nucleus of the solitary tract.

Little is known about the rapid action of VNS on neural activity in relevant brain structures. Previously, tVNS applications has shown to result in hypoactivation in the limbic system (amygdala, hippocampus and parahippocampal gyrus) [10]. Brain imaging studies suggest that activity of the limbic system is related to whether tinnitus is experienced as annoying or not [27]. Furthermore, invasive VNS has an antidepressant effect [28]. As various antidepressant approaches have shown to be effective in patients suffering from chronic tinnitus, it is assumed that antidepressant mechanisms influence tinnitus-related annoyance and handicap by modulation of emotion-regulatory brain structures. However, a further detailed understanding of the modulation of activity by vagus nerve stimulation is needed and may lead to the development of optimized stimulation protocols for patients suffering from tinnitus. Besides that, combining subjective tinnitus distress and symptom severity scores with objective tools, such as neuroimaging or electrophysiological studies (e.g. *f*MRI, EEG, or magnetoencephalogram) to assess the changes in neural activity, might be of additional value to evaluate the effect of neuromodulation treatments to relief tinnitus.

## Conclusion

In this systematic review we evaluated the effect of vagus nerve stimulation on tinnitus distress and symptom severity. Because of the high risk of bias, methodological limitations and low reporting quality of the included studies, no conclusions about the effectiveness of (t)VNS for reducing tinnitus distress or symptom severity could be drawn. To fully assess the effect of (t) VNS, a randomised controlled trial should be considered.

## Supporting information

**S1 Checklist. PRISMA 2009 checklist.**
(DOC)

## Author Contributions

**Conceptualization:** I. Stegeman, A. L. Smit.

**Data curation:** I. Stegeman, H. M. Velde, A. L. Smit.

**Formal analysis:** I. Stegeman, H. M. Velde, A. L. Smit.

**Investigation:** I. Stegeman, H. M. Velde, A. L. Smit.

**Methodology:** I. Stegeman, H. M. Velde, A. L. Smit.

**Project administration:** I. Stegeman, H. M. Velde, A. L. Smit.

**Resources:** P. A. J. T. Robe, R. J. Stokroos.

**Supervision:** I. Stegeman, A. L. Smit.

**Validation:** I. Stegeman, H. M. Velde, A. L. Smit.

**Writing – original draft:** I. Stegeman, H. M. Velde, P. A. J. T. Robe, R. J. Stokroos, A. L. Smit.

**Writing – review & editing:** I. Stegeman, H. M. Velde, P. A. J. T. Robe, R. J. Stokroos, A. L. Smit.

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
