## [Decision Letter · Decision Letter 0]

19 Aug 2020

PONE-D-20-12771

Tinnitus Treatment by Vagus Nerve Stimulation: A Systematic Review

PLOS ONE

Dear Dr. Stegeman,

Thank you for submitting your manuscript to PLOS ONE. After careful consideration, we feel that it has merit but does not fully meet PLOS ONE’s publication criteria as it currently stands. Therefore, we invite you to submit a revised version of the manuscript that addresses the points raised during the review process.

Please make sure to address all issues mentioned by the reviewers.

We look forward to receiving your revised manuscript.

Kind regards,

Sarah Michiels

Academic Editor

PLOS ONE

Journal Requirements:

2.PLOS requires an ORCID iD for the corresponding author in Editorial Manager on papers submitted after December 6th, 2016. Please ensure that you have an ORCID iD and that it is validated in Editorial Manager. To do this, go to ‘Update my Information’ (in the upper left-hand corner of the main menu), and click on the Fetch/Validate link next to the ORCID field. This will take you to the ORCID site and allow you to create a new iD or authenticate a pre-existing iD in Editorial Manager. Please see the following video for instructions on linking an ORCID iD to your Editorial Manager account: https://www.youtube.com/watch?v=_xcclfuvtxQ

3. Please ensure that you refer to Figure 1 in your text as, if accepted, production will need this reference to link the reader to the figure.

4. We note you have included a table to which you do not refer in the text of your manuscript. Please ensure that you refer to Table 3 and 4 in your text; if accepted, production will need this reference to link the reader to the Table.

<h1>** **</h1>

Reviewers' comments:

Reviewer's Responses to Questions

**Comments to the Author**

1. Is the manuscript technically sound, and do the data support the conclusions?

Reviewer #1: Partly

Reviewer #2: Partly

2. Has the statistical analysis been performed appropriately and rigorously? 

Reviewer #1: Yes

Reviewer #2: N/A

3. Have the authors made all data underlying the findings in their manuscript fully available?

Reviewer #1: Yes

Reviewer #2: Yes

4. Is the manuscript presented in an intelligible fashion and written in standard English?

Reviewer #1: Yes

Reviewer #2: Yes

5. Review Comments to the Author

Reviewer #1: There is a need for systematic review and meta-analysis of vagal nerve stimulation intervention and other forms of neuromodulation for tinnitus, and so the findings are a useful addition to the community. Generally, the methods and analysis have been carefully conducted and are consistent with good practice in systematic review methodology.

One of the recent protocols for a systematic review assessing treatment efficacy of a comparable intervention approach is that of neuromodulation for tinnitus published as a Cochrane review in 2015 (cochranelibrary.com/cdsr/doi/10.1002/14651858.CD011760/full). Treatments to be included in this review are vagal nerve stimulation (VNS), although it doesn’t seem to yet be published. Nevertheless, it is surprising that the authors don’t acknowledge this review in preparation, nor justify why another systematic review (which covers the same question) is needed.

The published Cochrane protocol on neuromodulation has as its primary outcome: Tinnitus symptom severity, and as Secondary outcomes: Generalised anxiety, Generalised depression, Generalised quality of life, Neurophysiological change (as measured by MEG or EEG) and Adverse effects. The current manuscript selects a different set of outcomes (tinnitus loudness, annoyance and distress), without explanation of why they are different, and of why this is justified. Failure to apply and report consistent sets of outcomes is a common reason why findings from different studies cannot be compared and evaluated. I would therefore recommend the authors to consider those outcomes sets used in recent Cochrane reviews in order to guide their own outcome selection.

A minor criticism is that the validated questionnaires of tinnitus listed in the manuscript are not all measures of tinnitus distress. Distress concerns the emotional aspects of tinnitus, whereas some of the questionnaires listed assess a range of symptoms including functional difficulties. Recommend the authors instead to use the same terminology as Cochrane ie ‘tinnitus symptom severity’ not ‘tinnitus distress’.

Another minor criticism is that the list of validated questionnaires includes the Tinnitus Acceptance Questionnaire (which is not recognised by recent Cochrane studies as a validated measure of symptom severity), and excludes the Tinnitus Severity Scale (which is recognised by recent Cochrane studies as a validated measure of symptom severity). Acceptance of tinnitus is usually addressed by CBT interventions and so its unclear why patients should be expected to improve on this measure.

Several of the authors should be familiar with this terminology ‘symptom severity’ not ‘distress’ and of the list of validated questionnnaires as they have recently co-authored a Cochrane review on betahistine for tinnitus.

Reviewer #2: this is a timely and valuable manuscript that may benefit from some changes:

1. from a neurobiological perspective VNS without sound pairing and with sound pairing can most likely not be mixed up, as they may be characterized by different mechanisms of action.

so a separate evaluation may be required.

the authors may want to quantify the changes. one theoretical way is to normalize the different scales to 100 and then evaluate the differences of the combined studies, analogous to what has been done in the pain field by Chakravarthy et al for burst stimulation in pain. this can be justified by the fact that most tinnitus questionnaires evaluate tinnitus related distress.

2. it may also be relevant to separate invasive vs non-invasive studies in a separate analysis: are invasive studies producing more pronounced effects or not?

6. PLOS authors have the option to publish the peer review history of their article (what does this mean?). If published, this will include your full peer review and any attached files.

Reviewer #1: No

Reviewer #2: **Yes: **Dirk De Ridder

---

## [Author Response · Author response to Decision Letter 0]

12 Nov 2020

Inge Stegeman

 University Medical Center Utrecht

 PO Box 85500

 NL-3508 GA Utrecht

 Email: i.stegeman@umcutrecht.nl

 Utrecht, sept, 2020

Dear Professor Michiels,

Please find attached our revised submission titled ‘Tinnitus Treatment by Vagus Nerve Stimulation: A systematic review’, which we would like you to consider for publication in PLOS ONE.

We thank the editor and reviewers for their careful consideration, and are pleased to have had the opportunity to revise our manuscript. Please find below a point-by-point response to all comments. 

We much appreciate the effort taken to reevaluate our manuscript and look forward to your reply. 

Sincerely on behalf of all authors,

Inge Stegeman, PhD

Corresponding author

 Reviewer 1 Response Manuscript changes

1 There is a need for systematic review and meta-analysis of vagal nerve stimulation intervention and other forms of neuromodulation for tinnitus, and so the findings are a useful addition to the community. Generally, the methods and analysis have been carefully conducted and are consistent with good practice in systematic review methodology. We thank the reviewer for this note. 

2 One of the recent protocols for a systematic review assessing treatment efficacy of a comparable intervention approach is that of neuromodulation for tinnitus published as a Cochrane review in 2015 (cochranelibrary.com/cdsr/doi/10.1002/14651858.CD011760/full). Treatments to be included in this review are vagal nerve stimulation (VNS), although it doesn’t seem to yet be published. Nevertheless, it is surprising that the authors don’t acknowledge this review in preparation, nor justify why another systematic review (which covers the same question) is needed. We thank the reviewer for this note. We have now mentioned the protocol in the discussion of our review. Page 22

3 The published Cochrane protocol on neuromodulation has as its primary outcome: Tinnitus symptom severity, and as Secondary outcomes: Generalised anxiety, Generalised depression, Generalised quality of life, Neurophysiological change (as measured by MEG or EEG) and Adverse effects. The current manuscript selects a different set of outcomes (tinnitus loudness, annoyance and distress), without explanation of why they are different, and of why this is justified. Failure to apply and report consistent sets of outcomes is a common reason why findings from different studies cannot be compared and evaluated. I would therefore recommend the authors to consider those outcomes sets used in recent Cochrane reviews in order to guide their own outcome selection.

 Consistency in reporting of outcomes is crucial for improving research in general, and is currently suboptimal, especially in tinnitus research. For tinnitus research though, a core-outcome set has not yet been developed. In 2015 in the COMMIT’ID trial this issue was addressed, and core domain sets were created. This provided a guideline for new studies to create their outcome sets more uniformly for the future. For several groups of tinnitus therapies different outcomes were set. Neuromodulation therapies were not taken into account. 

When designing this systematic review we aimed to focus on primarily tinnitus outcomes of VNS therapy. Because of the heterogeneity in outcomes and outcome measures historically in the field of tinnitus research we wanted to take into account the full range of tinnitus outcomes described in the past. To underline the importance of sound methodology we added this to the discussion chapter. Page 22. 

4 A minor criticism is that the validated questionnaires of tinnitus listed in the manuscript are not all measures of tinnitus distress. Distress concerns the emotional aspects of tinnitus, whereas some of the questionnaires listed assess a range of symptoms including functional difficulties. Recommend the authors instead to use the same terminology as Cochrane ie ‘tinnitus symptom severity’ not ‘tinnitus distress’. We thank the reviewer for this improvement in our manuscript, which we have implemented. 

5 Another minor criticism is that the list of validated questionnaires includes the Tinnitus Acceptance Questionnaire (which is not recognized by recent Cochrane studies as a validated measure of symptom severity), and excludes the Tinnitus Severity Scale (which is recognized by recent Cochrane studies as a validated measure of symptom severity). Acceptance of tinnitus is usually addressed by CBT interventions and so its unclear why patients should be expected to improve on this measure.

Several of the authors should be familiar with this terminology ‘symptom severity’ not ‘distress’ and of the list of validated questionnaires as they have recently co-authored a Cochrane review on betahistine for tinnitus. We acknowledge the note of the reviewer concerning the Tinnitus Acceptance Questionnaire, acceptance is not the main objective of VNS implantation. Nor do we expect that implantation will increase acceptance. Therefore we have, after some discussion, deleted TAQ from the outcome list. 

We did not include the tinnitus severity scale in the outcomes, which we have now included. No studies reported on this scale. 

 Reviewer 2 (dirk de ridder) Response Manuscript changes

1. this is a timely and valuable manuscript that may benefit from some changes

 We would like to thank dr. De Ridder for his comments 

2. from a neurobiological perspective VNS without sound pairing and with sound pairing can most likely not be mixed up, as they may be characterized by different mechanisms of action.

so separate evaluation may be required. We have now sub-divided the table in a part with and without sound pairing. 

3. the authors may want to quantify the changes. one theoretical way is to normalize the different scales to 100 and then evaluate the differences of the combined studies, analogous to what has been done in the pain field by Chakravarthy et al for burst stimulation in pain. this can be justified by the fact that most tinnitus questionnaires evaluate tinnitus related distress.

it may also be relevant to separate invasive vs non-invasive studies in a separate analysis: are invasive studies producing more pronounced effects or not? We do agree with dr. De Ridder that summarizing results in one outcome would be beneficial. Though because of the large heterogeneity in included patients and questionnaires, we do think that normalizing the different scales to 100 would not provide an accurate view of the results. We are afraid that it gives the impression of pooling and homogeneity. 

We think that it is a too early to draw conclusions about differences, also not for subgroups of therapies. We have now added this statement to the discussion. 

‘However, given the aforementioned limitations, no strong conclusions could be drawn from these data on the effect of (t)VNS with or without sound therapy, nor about differences between implanted or transcutaneous VNS.’

---

## [Decision Letter · Decision Letter 1]

4 Feb 2021

Tinnitus Treatment by Vagus Nerve Stimulation: A Systematic Review

PONE-D-20-12771R1

Dear Dr. Stegeman,

We’re pleased to inform you that your manuscript has been judged scientifically suitable for publication and will be formally accepted for publication once it meets all outstanding technical requirements.

Kind regards,

Sarah Michiels

Academic Editor

PLOS ONE

Additional Editor Comments (optional):

Reviewers' comments:

Reviewer's Responses to Questions

**Comments to the Author**

1. If the authors have adequately addressed your comments raised in a previous round of review and you feel that this manuscript is now acceptable for publication, you may indicate that here to bypass the “Comments to the Author” section, enter your conflict of interest statement in the “Confidential to Editor” section, and submit your "Accept" recommendation.

Reviewer #2: All comments have been addressed

2. Is the manuscript technically sound, and do the data support the conclusions?

Reviewer #2: Yes

3. Has the statistical analysis been performed appropriately and rigorously? 

Reviewer #2: I Don't Know

4. Have the authors made all data underlying the findings in their manuscript fully available?

Reviewer #2: Yes

5. Is the manuscript presented in an intelligible fashion and written in standard English?

Reviewer #2: Yes

6. Review Comments to the Author

Reviewer #2: no further questions

no further questions

no further questions

no further questions

no further questions

7. PLOS authors have the option to publish the peer review history of their article (what does this mean?). If published, this will include your full peer review and any attached files.

Reviewer #2: **Yes: **Dirk De Ridder

---

## [Editor Report · Acceptance letter]

3 Mar 2021

PONE-D-20-12771R1 

Tinnitus Treatment by Vagus Nerve Stimulation: A Systematic Review 

Dear Dr. Stegeman:

I'm pleased to inform you that your manuscript has been deemed suitable for publication in PLOS ONE. Congratulations! Your manuscript is now with our production department. 

Kind regards, 

on behalf of

Prof. Sarah Michiels 

Academic Editor

PLOS ONE